

# Life-expectancy changes during the COVID-19 pandemic from 2019–2021: estimates from Japan, a country with low pandemic impact

Mst Sirajum Munira, Yuta Okada and Hiroshi Nishiura

School of Public Health, Kyoto University, Kyoto, Japan

## ABSTRACT

**Background:** The COVID-19 pandemic had a low impact on Japan in 2020, but the size of the epidemic increased considerably there in 2021. This study made a statistical analysis of life expectancy changes up to the end of 2021 in Japan.

**Objective:** We aimed to estimate changes in life expectancy from 2019 to 2021 associated with the COVID-19 pandemic. To do so, we decomposed the life expectancy changes from 2020–2021 into age-specific and cause of death-specific contributions.

**Methods:** We used the absolute number of deaths by age and prefecture in Japan to calculate life expectancy from 2019–21 at both national and prefectural levels, and also examined the correlation between life expectancy gap and annual number of COVID-19 cases, total person-days spent in intensive care, and documented deaths due to COVID-19. We used the Arriaga decomposition method to decompose national life expectancy changes from 2020 to 2021 into age and cause of death components.

**Results:** From 2019–2020, Japan's national level life expectancy across the entire population was extended by 0.24 years. From 2020–2021, it shortened by 0.15 years. The life expectancy shortened more among women (0.15 years) than men (0.12 years). There was significant heterogeneity in life expectancy changes from 2020–2021 by prefecture. It ranged from the maximum shortening of 0.57 years in Tottori prefecture to the maximum extension of 0.23 years in Fukui. The regression analysis revealed the negative correlation between the life expectancy change and burden of COVID-19 at prefectural level. The decomposition of life expectancy changes at birth from 2020–2021 showed that losses in life expectancy were largely attributable to the mortality of the population over 70 years old. Changes in life expectancy among infants and working-age adults mostly contributed to lengthening overall life expectancy. Among leading major causes of death, deaths due to neoplastic tumor and cardiovascular diseases contributed to shortening life expectancy, whereas respiratory diseases did not.

**Conclusion:** The decades-long increasing trend in life expectancy was suspended by the COVID-19 pandemic. However, life expectancy changes from 2019–2020 and 2020–2021 were small in Japan. This may be attributable to the small epidemiological impact of COVID-19 during this time period, but nonetheless, the negative impact of COVID-19 on life expectancy was indicated in the present study. The chance of death accelerated in older people in 2021, but a smaller number of deaths than usual

Corresponding author
Hiroshi Nishiura,
nishiurah@gmail.com

among infants and working age adults contributed to extended life expectancy, and the change in the cause of death structure under the COVID-19 pandemic also significantly contributed to shortening life expectancy.

## INTRODUCTION

COVID-19 emerged in Wuhan, China in November 2019, and rapidly spread to become a pandemic. To suppress the significant morbidity and mortality impact of COVID-19, governments around the world introduced various public health and social measures, especially before the introduction of vaccines. Despite this action, COVID-19 had led to 6.84 million deaths worldwide by 4 February 2023 on the basis of official reports from countries worldwide, even in high-income countries such as the United States where COVID-19 was the third leading cause of death in 2021 (*Ahmad, Cisewski & Anderson, 2022*). The excess mortality studies suggest that the mortality impact of COVID-19 is even more significant and the extent of impact varies across regions (*Bilal et al., 2021*; *Wang et al., 2022*; *Msemburi et al., 2022*). The World Health Organization (WHO) estimated that even between 1 January 2020 and 31 December 2021, the total number of excess deaths globally reached about 14.9 million with substantial contributions from the United States and India (*Wang et al., 2022*; *Msemburi et al., 2022*).

In addition to excess mortality estimates, life expectancy is another important figure that suggests the mortality impact of COVID-19 including in high income countries (*Woolf, Masters & Aron, 2021*, *2022*; *Aburto et al., 2022*). In the United States, life expectancy at birth was shortened by 1.87 years from 2019 to 2020, and a similar tendency is expected in other high-income countries, due to a surge in COVID-19 cases during the pre-vaccination period (*Woolf, Masters & Aron, 2022*). The life expectancy changes from 2021–22 in the United States is yet to be officially reported, but additional shortening of life expectancy is anticipated due to the continued impact of the pandemic.

The mortality impact of the pandemic is not only caused directly by COVID-19 as an acute respiratory entity, but also by clinical complications of COVID-19 such as cardiovascular events or mortality impact *via* other complications are gaining growing attention recently (*Knight et al., 2022*; *Raisi-Estabragh et al., 2023*; *Lee et al., 2023*). Mortalities that are indirectly caused by COVID-19, in most cases, are not necessarily reflected as COVID-19 deaths in the vital statistics.

In Japan, the COVID-19 epidemic size has been relatively small compared with other high-income countries, at least up to the end of the primary vaccination series at the end of 2021 (*Ministry of Health, Labour and Welfare, 2023a*). Until the end of 2021, only less than 5% of the total population experienced COVID-19 infection (*Ministry of Health, Labour and Welfare, 2023a*), and excess mortality of 13,374 was estimated to have occurred from January 2020 to December 2021 (*Excess & Exiguous Deaths Dashboard in Japan, 2023*), which is a relatively small figure compared with over a million total annual deaths in Japan.

Several studies in Japan also suggest that mortality or excess death was relatively controlled in Japan (*Nomura et al., 2022b*, *2022a*; *Onozuka et al., 2022*).

As is suggested from findings on mortality under the COVID-19 pandemic in Japan, the life expectancy in Japan did not significantly change in 2020, but a slight shortening in life expectancy was observed from 2020 to 2021 (*National Institute of Population and Social Security Research, 2023*). Considering the small excess mortality in Japan, further detailed investigation of how this change in life expectancy occurred in Japan under the COVID-19 pandemic is a key public health interest.

This study aimed to estimate the life expectancy changes in Japan throughout the COVID-19 pandemic from 2019–2021, at both national and prefectural levels. We also aimed to decompose the life expectancy changes over the period into age-specific and cause-of-death-specific contributions.

## MATERIALS AND METHODS

### Demographic and epidemiological datasets

Data on deaths and exposure-to-risk populations from 2019–2021 were retrieved from the Japanese Mortality Database (*National Institute of Population and Social Security Research, 2023*). Data on deaths by causes and by age group from 2020–2021 was obtained from Vital Statistics provided by the (*Ministry of Health, Labour and Welfare, 2022b*, *2022c*). We used the International Statistical Classification of Diseases and Related Health Problems 10th Revision (ICD-10) to group deaths into top nine major cause categories (based on death by causes in 2021) and the remainder to make a total of 10 groups. Finally, we obtained data on the daily number of COVID-19 cases and severe cases in 2021 by prefecture from the Ministry of Health, Labour and Welfare's COVID-19 Open Data (*Ministry of Health, Labour and Welfare, 2023a*).

### Calculation of life expectancy

The life table for the period was computed using the methods adopted by the Japan Mortality Database with slight modification (*National Institute of Population and Social Security Research, 2022*). The age group-specific mortality rate $m_x$, where $x$ represents the age group of interest, was calculated using the exposure-to-risk population. For $a_x$, the average length of time to death in deceased individuals in age group $x$, we assumed the same value for the 2020 and 2021 life tables, because the $a_x$ for 2021 was not available from the Japanese Mortality Database at the time of our analysis.

Using $m_x$ and $a_x$, the probability that people in age group $x$ do not survive to age group $x + 1$, $q_x$, can be obtained using the following formula:

$$q_x = \frac{m_x \, w_x}{1 + (w_x - a_x) \, m_x},$$

where $w_x$ is the length of discrete interval of age group $x$. Then, using $q_x$, the number of survivors from birth to each exact age is obtained as follows:

$$l_0 = 100,000,$$

$$l_{x+1} = l_x(1 - q_x),$$

for age group $x = 0, 1-4, 5-9, \ldots 110$ and older (in years). The number of deaths in age group $x$ is:

$$d_x = l_x q_x.$$

Given $l_x$, $a_x$, and $d_x$, the term $L_x$ can be obtained as the sum of person-years spent in age group $x$ as follows:

$$L_x = w_x l_{x+1} + a_x d_x,$$

and the remaining person-years of life for those in age group $x$, $T_x$, is written as:

$$T_x = \sum_{i=x}^{110+} L_i.$$

The life expectancy of age group $x$, $e_x$, is then calculated as follows:

$$e_x = \frac{T_x}{l_x}$$

For the rest of this article, the life expectancy is calculated at age zero (at the time of birth).

Calculating the life tables for the years 2019–2021, we computed life expectancy changes for Japan as a whole, as well as by prefecture from 2019–2020 and 2020–2021. We then explored the correlation between this annual change in life expectancy and the cumulative number of COVID-19 cases, annual number of person-days receiving intensive care, and total number of documented deaths by COVID-19 by prefecture using a linear regression method and employing the expectancy gap as the dependent variable. As the impact of COVID-19 in Japan was relatively limited, the life expectancy change was also compared against infant mortality rate and the natural birth rate against total population in 2021 by prefecture (*Ministry of Health, Labour and Welfare, 2022b*).

## Decomposing the life expectancy changes

To understand the age-specific and cause-specific contribution to the life expectancy gap between 2020 and 2021 for the whole of Japan, we used the Arriaga decomposition method (*Arriaga, 1984*; *Chisumpa & Odimegwu, 2018*). The total contribution of age group $x$ to the life-expectancy change (in years), denoted as $C_x$, can be described as:

$$C_x = \left[ \frac{l_x^{2020}}{l_0} \left( \frac{L_x^{2021}}{l_x^{2021}} - \frac{L_x^{2020}}{l_x^{2020}} \right) \right] + \left[ \frac{T_{x+1}^{2021}}{l_0} \left( \frac{l_x^{2020}}{l_x^{2021}} - \frac{l_{x+1}^{2020}}{l_{x+1}^{2021}} \right) \right], \tag{1}$$

where the first term is the direct contribution, and the second term in brackets is the indirect or interaction contribution to the difference in life expectancy attributable to the corresponding age group. $L_x$ is the number of person-years lived within age group $x$, and $T_{x+1}$ is the total number of person-years lived within age groups $x + 1$ and beyond. The last age group only has a direct effect because there is no later age group for any

**Table 1  Life expectancy at birth from 2019 to 2021, Japan.**

| Population | 2019 | 2020 | 2021 |
|---|---|---|---|
| Entire Japan | 84.49 | 84.73 | 84.58 |
| Male | 81.41 | 81.61 | 81.49 |
| Female | 87.47 | 87.76 | 87.61 |

**Note:**
Calculated number of years that people can survive as a mean is shown (unit is years). See Tables S1–S3 for the entire life tables.

indirect or interaction effect. The sum of contributions from all age groups should equal the total life expectancy change in years. We used this method for the age groups up to 100+ years old, because the cause of death data for those aged 100–104, 105–109 and 110 years or more were not available. We therefore recalculated the life table for age groups from 0 to 100 years or more to allow decomposition.

We then decomposed the life expectancy change by cause of death. We assumed that the contribution of the change in mortality from each cause $i$ in age group $x$ to the change in the number of person-years in age group $x$ is proportional to the contribution of this cause to the change in the central (overall) mortality rate in age group $x$. Under this assumption, $C_x$ can be decomposed into proportions attributable to each cause of death $i$, which were denoted by $C_x^i$, using the following formula:

$$C_x^i = C_x \left[ \frac{R_x^{i,2021} m_x^{2021} - R_x^{i,2020} m_x^{2020}}{m_x^{2021} - m_x^{2020}} \right], \qquad (2)$$

where $R_x^i$ is the proportion of deaths in age group $x$ due to cause $i$, and $m_x$ is the all-cause mortality rate in age group $x$.

## Statistical and numerical analysis
All the analyses were computed using R version 4.2.1 (*R Core Team, 2023*).

## Data sharing statement
Cause of death data are available as Tables S14–S22.

## Ethical considerations
This study used publicly available information. Ethical approval was therefore not required.

## RESULTS
Table 1 summarizes the estimated life expectancies at birth in 2019, 2020 and 2021, computed by sex. From 2019 to 2020, the life expectancy grew by 0.24 years, from 84.49 years to 84.73 years. However, from 2020 to 2021, it decreased by 0.15 years from 84.73 years to 84.58 years. The male life expectancy grew by 0.20 years from 2019 to 2020 and decreased by 0.12 years from 2020 to 2021. The female life expectancy grew by 0.29 years from 2019 to 2020 and decreased by 0.15 years from 2020 to 2021 (See Fig. S1 for comparison between actual post-COVID-19 life expectancy and pre-COVID-19 life

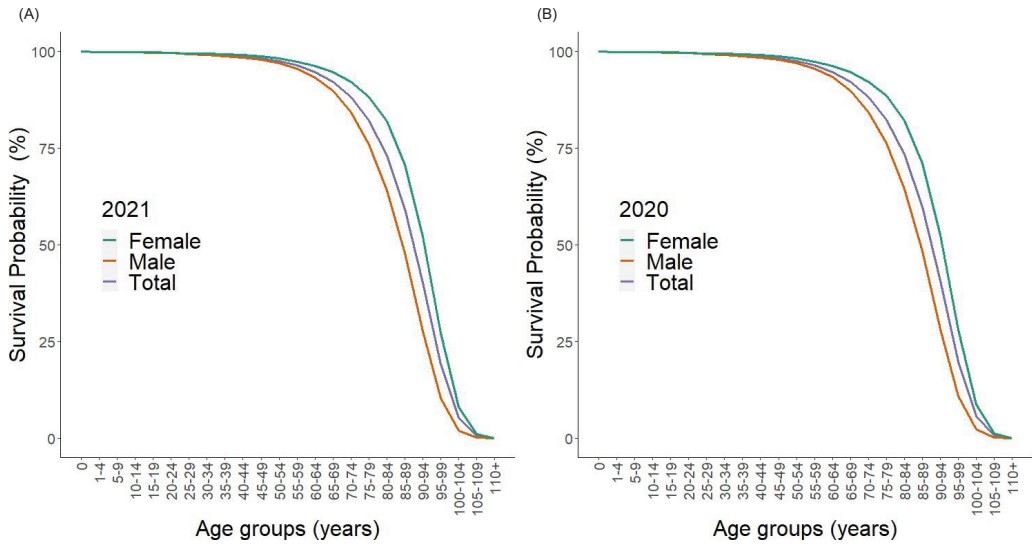

**Figure 1 Survival curves of the population in Japan, 2020 and 2021.** Survival curves are shown, as derived from the $l_x$ in the life table. Left-hand side shows survivorship at the end of 2021, and right-hand side 2020. At age zero, 100% are alive. The blue, green, and red lines represent the $l_x$ for the total, male, and female populations. There is no visually recognizable difference between 2020 and 2021.

expectancy trend. Also, see Tables S1–S9 for life table and life expectancy estimate by year, sex, and place.).

Figure 1 shows the survival curve based on the life table for the total population and by sex in Japan for 2020 and 2021 (calculated as on 31 December of each year). The curves correspond to the survival percentage $s_x$ since birth, as a function of age $x$, which can be written as:

$$s_x = \frac{l_x}{l_0}$$

In line with expectations, women tended to live longer than men, and there was no major change seen in the survival curve from 2020–2021.

Figure 2 shows life expectancy changes of the total population by prefecture from 2019–2020 and 2020–2021. From 2020–2021, the change ranged from the maximum decrease of 0.57 years in Tottori to the maximum growth of 0.23 years in Fukui, and shortening of life expectancy was observed in 34 out of 47 total prefectures (Fig. 2A). This finding contrast to the change from 2019–2020, *i.e.*, only three prefectures experienced a slight shortening in life expectancy and most prefectures saw an increase in life expectancy of less than 1 year (Fig. 2B).

Prefecture that experienced the largest decrease in male life expectancy in 2021 was Gunma (0.62 years) (Table S10), and the largest decrease in female life expectancy were observed in Oita (0.63 years). The gap in life expectancy between men and women decreased in 27 prefectures during the pandemic. The greatest decrease in gap was in Oita (0.68 years), and the greatest increase was in Kochi (0.79 years) (Table S10).

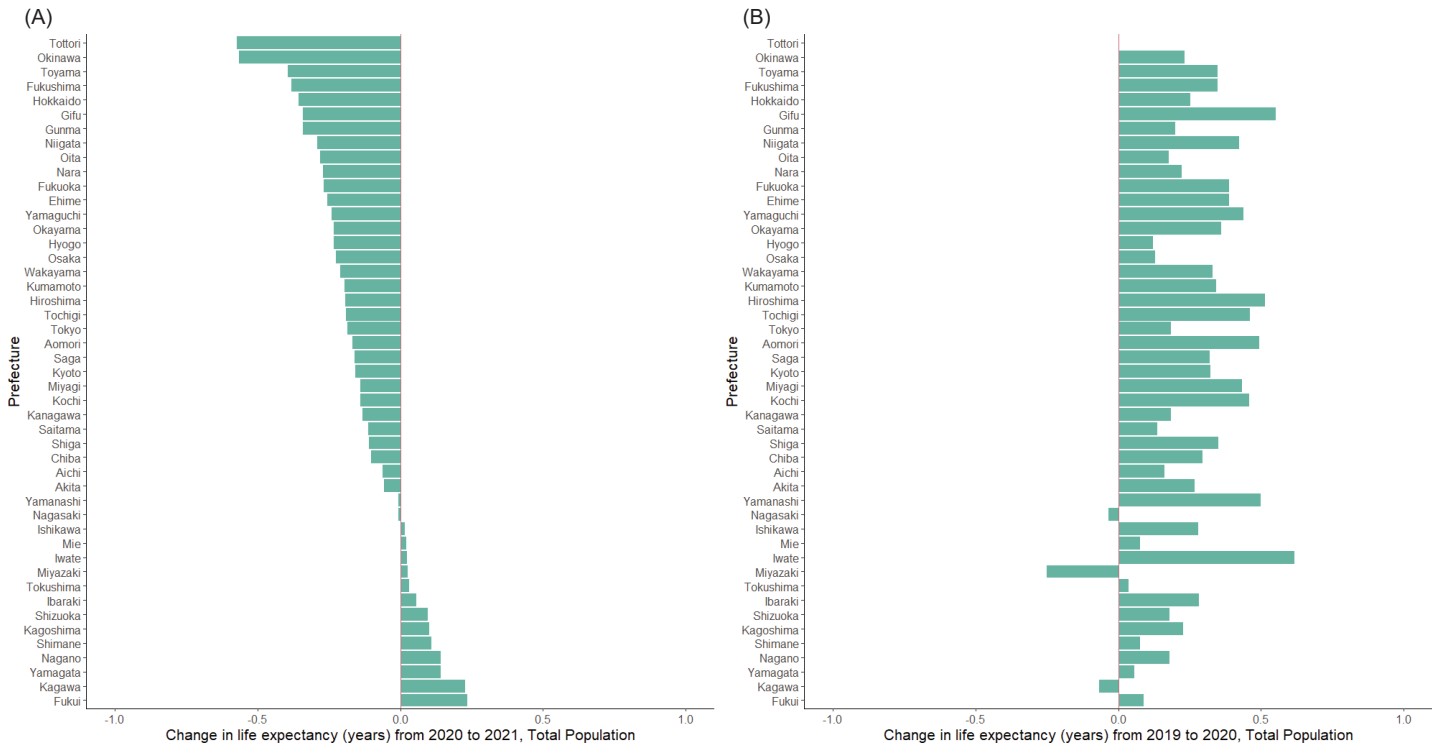

**Figure 2 Life expectancy changes by prefecture in Japan, from 2019–2020 and 2020–2021.** Change in the life expectancy at birth for the total population was calculated from (A) 2020–2021 and (B) 2019–2020. In panel A, prefectures are ordered by the life expectancy change. The order in panel B follows that in A.

Figure 3 shows the correlation between life expectancy change and the cumulative number of cases, annual number of person-days receiving intensive care, or cumulative number of documented deaths by COVID-19. The linear regression showed negative correlation between expectancy gap and these variables, implying the possible negative impact of COVID-19 on life expectancy (Table 2). Similar finding was obtained when using prefectural infant mortality rate, but the prefectural birth rate did not show any significant correlation with the expectancy gap (Figs. S2).

Figure 4 shows the age decomposition of the life expectancy gap for 2020–2021 (A) and 2019–2020 (B) for the whole population. Changes in the population aged 70 years or older were the major contributors to the lower life expectancy from 2020–2021, which contrasts to that from 2019–2020. Another key finding is that the negative impact from the age range 15–29 years in 2019–2020 have shrink significantly in 2020–2021. Changes in life expectancy among infants up to age 1 year still contributed to extended life expectancy in 2020–2021, although the extent is small compared with that in 2019–2020.

The decomposition by gender revealed the same tendency (Table S13). Figure 5 shows the decomposition of life expectancy change by major cause of death from 2020–2021 (A) and 2019–2020 (B). Neoplasm and cardiovascular diseases shortened the life expectancy (−0.028 years and −0.041 years, respectively) whereas respiratory diseases contributed to slight lengthening (0.007 years). Similar patterns were observed when cause-of-death-specific decomposition was performed by sex (Table S12).

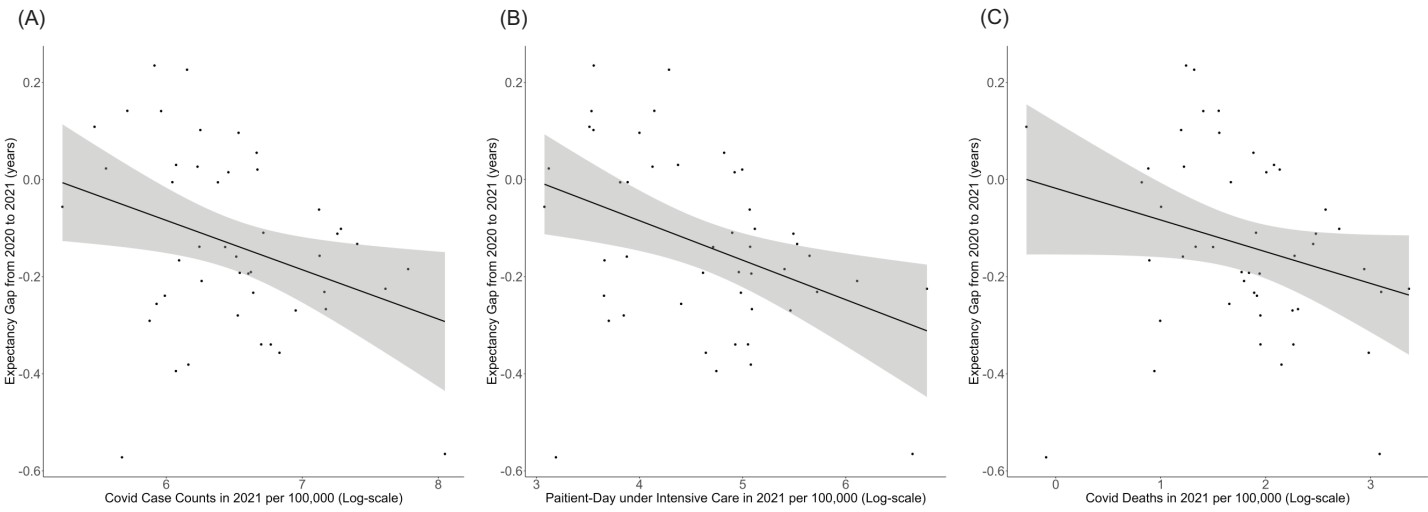

**Figure 3 (A–C) Correlation between life expectancy change from 2020–2021 and the annual cumulative number of COVID-19 cases per 100,000, patient-day under intensive care per 100,000, and cumulative number of COVID-19 deaths in 2021.** Each dot represents the estimate in a single prefecture (there are 47 dots). The line shows the best fit from linear regression, and the 95% confidence intervals are shown by shading.                                         

**Table 2 Life expectancy gap and COVID-19 burden in 2021: linear regression analysis on prefectural data.**

| COVID-19 data (in log scale) | Coefficient (95% CI) | Intercept (95% CI) |
|---|---|---|
| Annual cases | −0.102 [−0.185 to −0.0181] | −0.645 [−1.067 to −0.222] |
| Annual person-day in intensive-care | −41.44 [−71.09 to −11.79] | −0.072 [−0.139 to −0.005] |
| Annual death | −0.065 [−0.134 to 0.003] | −0.770 [−1.437 to −0.103] |

## DISCUSSION

This study explored how the life expectancy in Japan was affected by the COVID-19 pandemic from 2020–2021, where the epidemic size was relatively small from a global perspective at least until the end of 2021. Our life table analysis of 2019, 2020 and 2021 showed that there was a slight extension of life expectancy in 2020, occurring in 44 out of 47 prefectures. The epidemic increased in size in 2021, and the life expectancy was slightly shortened in 34 out of 47 prefectures. We also found substantial heterogeneities in life expectancy change by age, sex and place. To our knowledge, this is the first study to thoroughly quantify the death dynamics from 2020–2021 in Japan in the presence of COVID-19, analyzing the data from various angles including prefectural analysis and decomposition analysis by age and causes of death.

The available life expectancy data from 1947 onwards at the Japanese Mortality Database shows that life expectancy in Japan has basically been maintained the continuously growing trend since 1947, shortly after World War II. Our study has shown that life expectancy in 2021 deviated from the trend over 20 years before the COVID-19 era (Fig. S1). However, from a global point of view, the magnitude of the decrease was smaller than those reported by other high-income countries (*Bilal et al., 2021*; *Aburto et al.,*

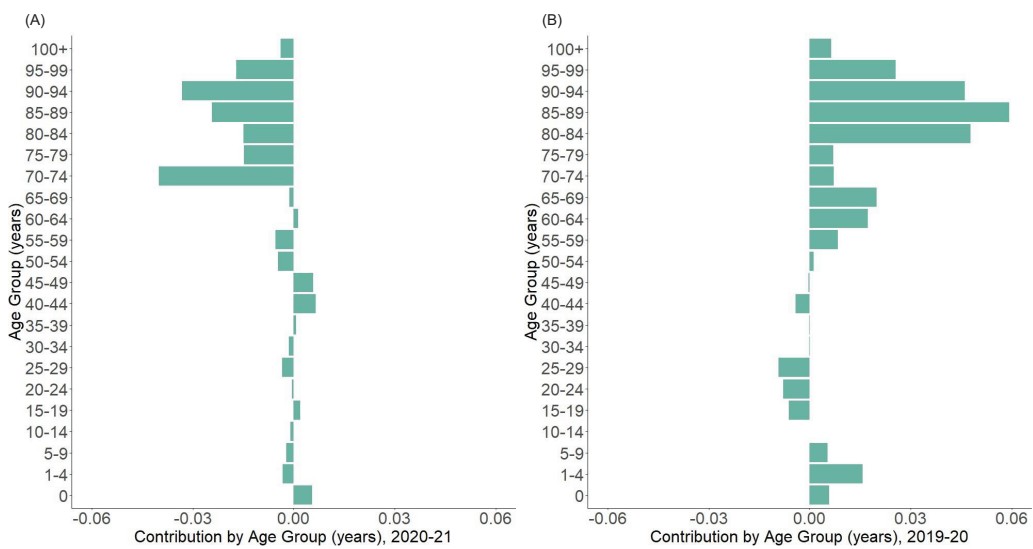

**Figure 4 Arriaga decomposition of life expectancy change from 2020–2021 by age group.**
(A) Decomposed contribution by age from 2020 to 2021, showing the negative contribution of population over Age 70 to the life expectancy change from 2020 to 2021, compared with younger age groups. (B) Decomposed contribution by age from 2019 to 2020, showing the positive contribution of population over Age 50, which is contrary to what is observed from 2020 to 2021.

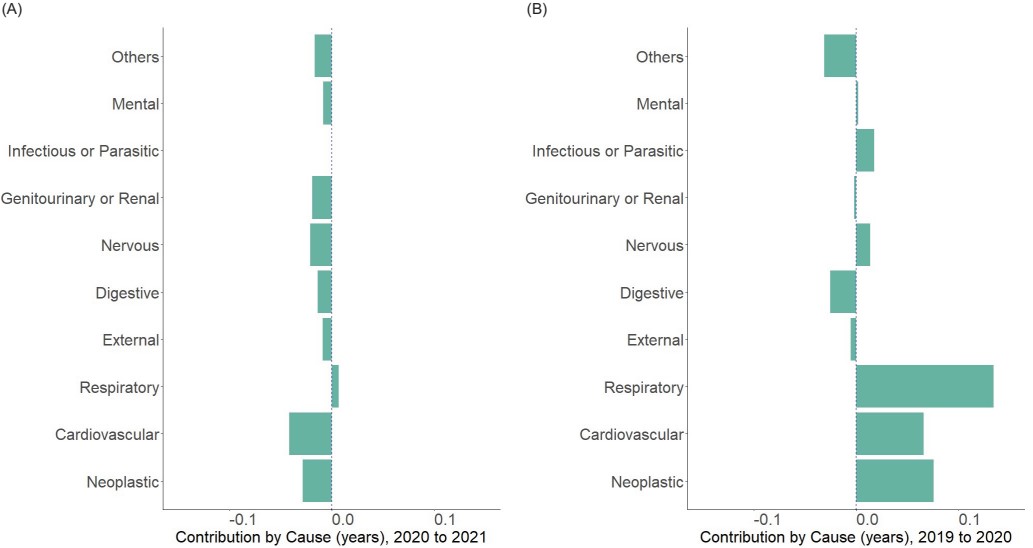

**Figure 5 Arriaga decomposition of life expectancy change from 2020–2021 by major causes of death.**
(A) Decomposed contribution by major causes of death from 2020 to 2021, showing the relatively large contribution of neoplastic and cardiovascular diseases from 2020 to 2021, compared with other major death causes. (B) Decomposed contribution by major causes of death from 2019 to 2020, showing positive contributions from neoplastic, cardiovascular, and respiratory causes, which is contrary to the result in (A).

*2022*; *Woolf, Masters & Aron, 2022*; *Schwandt et al., 2022*) and the absence of abrupt shortening in life expectancy in Japan is notable. Our analysis on the relationship between life expectancy change and epidemiological burden of COVID-19 suggests that even the

relatively small epidemic size of COVID-19 in Japan has led to a detectable mortality impact. Because the cumulative confirmed number of COVID-19 cases in Japan at the end of 2021 was only 1.37% of the overall population (*Ministry of Health, Labour and Welfare, 2023a*), this may suggest that we will see huge negative impact of COVID-19 on life expectancy in Japan in 2022 or later considering that the epidemic size has dramatically grown. Another finding of interest is also the negative correlation between life expectancy change and infant mortality rate. The underlying mechanism or causality cannot be inferred from our results, but this might be to some extent an indicator of healthcare capacity or healthcare seeking behavior within each prefecture at the baseline regardless of COVID-19. This issue should be an issue to be further investigated in future studies.

In addition to the impact of COVID-19 on life expectancy change from 2020 to 2021, our decomposition analysis of life expectancy changes from 2020–2021 in comparison with that from 2019–2020 provided useful insights into the mechanisms of death by age and cause of death. Changing life expectancy among older people certainly contributed to the shorter life expectancy in 2021, which was contrary to that in 2020 and is consistent with the clinical nature of COVID-19 that causes more mortality impact in the elderly. Younger age groups seem to have a relatively small contribution to life expectancy shortening from 2020–2021, but compared with 2019–2020, negative impact was also observed.

Our decomposition analysis of life expectancy changes by major causes of death revealed that neoplastic and cardiovascular diseases were the two highest contributing categories to shortening of life expectancy in 2021. Compared with 2019–2020, the drastic decrease of contributions from cardiovascular, neoplastic and respiratory diseases was notable. In addition to the negative impact of COVID-19 and its complications, decreased or limited access to healthcare during the pandemic may have also been the underlying cause, because significant decrease and delays in hospital visits have been reported in Japan (*Yamaguchi et al., 2022*; *Terashima et al., 2022*; *Yoshida et al., 2022*; *Ii & Watanabe, 2022*). Our additional analysis between the life expectancy changes in 2020–2021 and hospital visit, hospital stay (*Ministry of Health, Labour and Welfare, 2022a*) and cancer mortality below age 75 per 100,000 (*Foundation for Promotion of Cancer Research, 2023*) did not reveal significant findings, although mild tendency of negative correlation between cancer mortality and life expectancy was observed (Figs. S3). Further investigation into the possible causal structure regarding this issue is warranted.

Future studies will be needed to explore the demographic change after 2021. Even though it is likely that the life expectancy shortening and excess mortality in Japan occurred mainly in the elderly population, still, there is significant room for COVID-19 infections in these older populations to cause a substantial number of deaths following the reopening of society during the post-vaccination period for COVID-19, from 2022 onwards. Similarly, the disease structure may change from 2022, because a substantial number of infants and older middle-aged people with underlying comorbidity survived from 2020–2021 owing to the infection control measures against COVID-19. We plan to clarify these elements in a future study.

This study had several limitations that need to be discussed. First, we were not able to explore more about the relationship between life expectancy change and COVID-19, because we were not able to access prefectural data on the cause of death. At the national level, we have been able to provide useful insights, but additional analysis would have helped to understand geographically differing patterns of death. Second, data on COVID-19 in this study were prone to ascertainment bias, *i.e.*, cases may represent only the tip of the pandemic iceberg. However, we are confident that we have been able to examine the relationship between life expectancy change and person-day of COVID-19 cases requiring intensive care, possibly attenuating the impact of diminished diagnostic coverage. Third, the coverage of death registration in Japan, which is reported to be 90–99% by the United Nations (*United Nations Statistics Division, 2023*), might also be considered as a limitation of our study. However, it is unlikely that deaths in the older population, which have the greatest impact on life expectancy, were missing from the official statistics on a large scale.

Even considering these limitations, we believe that our study provides important insights into the changes in life expectancy in a country with a relatively small COVID-19 epidemic. We have shown that at the national level, life expectancy was not greatly shortened in 2020 and only slightly and partly shortened in 2021. This may be largely attributable to the relatively small size of the COVID-19 pandemic in Japan, but still the negative impact of COVID-19 on life expectancy was observed. Our results lay the foundation for further demographic analyses of deaths in Japan.

## CONCLUSIONS

This study explored the life expectancy in Japan during the course of the COVID-19 pandemic, from 2020–2021. Computing life tables at the end of years 2019, 2020 and 2021, we found a slight extension of life expectancy in 2020, occurring in 44 out of 47 prefectures. The epidemic increased in size in 2021, and the life expectancy was slightly shortened in 33 out of 47 prefectures. We also found substantial heterogeneities in life expectancy change by age, sex and place. We believe that our study provides important insights into the changes in life expectancy in a country with a relatively small COVID-19 epidemic.

## ACKNOWLEDGEMENTS

We thank Melissa Leffler, MBA from Edanz for editing a draft of this manuscript.

### Funding

Hiroshi Nishiura received funding from Health and Labour Sciences Research Grants (20CA2024, 20HA2007, 21HB1002 and 21HA2016); the Japan Agency for Medical Research and Development (AMED; JP20fk0108140, JP20fk0108535, JP21fk0108612 and JP 23fk0108685); the Japan Society for the Promotion of Science KAKENHI (21H03198 and 22K19670); the Environment Research and Technology Development Fund (JPMEERF20S11804) of the Environmental Restoration and Conservation Agency of Japan; the Japan Science and Technology Agency SICORP Program (JPMJSC20U3 and

JPMJSC2105), and the RISTEX program for Science of Science, Technology and Innovation Policy (JPMJRS22B4). The funders had no role in study design, data collection and analysis, decision to publish, or preparation of the manuscript.

## Grant Disclosures
The following grant information was disclosed by the authors:
Health and Labour Sciences Research Grants: 20CA2024, 20HA2007, 21HB1002 and 21HA2016.
Japan Agency for Medical Research and Development: JP20fk0108140, JP20fk0108535, JP21fk0108612 and JP 23fk0108685.
Japan Society for the Promotion of Science KAKENHI: 21H03198 and 22K19670.
Environment Research and Technology Development Fund: JPMEERF20S11804.
Environmental Restoration and Conservation Agency of Japan.
Japan Science and Technology Agency SICORP Program: JPMJSC20U3 and JPMJSC2105.
RISTEX program for Science of Science.
Technology and Innovation Policy: JPMJRS22B4.

## Competing Interests
Hiroshi Nishiura is an Academic Editor for PeerJ. Otherwise, the authors declare that they have no competing interests.

## Author Contributions
- Mst Sirajum Munira performed the experiments, analyzed the data, prepared figures and/or tables, authored or reviewed drafts of the article, and approved the final draft.
- Yuta Okada conceived and designed the experiments, performed the experiments, analyzed the data, prepared figures and/or tables, authored or reviewed drafts of the article, and approved the final draft.
- Hiroshi Nishiura conceived and designed the experiments, performed the experiments, authored or reviewed drafts of the article, and approved the final draft.

## Data Availability
The cause of death data are available in the Supplemental Tables.

## Supplemental Information
Supplemental information for this article can be found online at http://dx.doi.org/10.7717/peerj.15784#supplemental-information.

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
