# Peer review of "Life-expectancy changes during the COVID-19 pandemic from 2019–2021: estimates from Japan, a country with low pandemic impact"

_PeerJ, doi:10.7717/peerj.15784_

## Round 0.1 · original submission · Major Revisions

Though the reviewers and myself found the manuscript interesting, there is a handful of additional analyses that would be necessary to improve the quality of the manuscript and its potential impact. Details are provided in the reviewers' comments. Please, make sure to address all reviewers' comments as the revised manuscript will be returned to reviewers for further evaluation.

Reviewer 1 ·

Basic reporting

No comments

Experimental design

No comments

Validity of the findings

No comment

Additional comments

1. The authors focus on two years, 2020 and 2021, and there is a slight increase and decrease in the life expectancy the two years. If their purpose is to assess losses in life expectancy during the pandemic, they are supposed to compare life expectancy under a counterfactual scenario where there were no pandemic. Indeed, life expectancy has been increasing over time and without pandemic we expect further increase both in 2020 and 2021. In the current analysis, they focus on a decrease in life expectancy in 2021 compared to the actual value of the previous year 2020, but what it should be compared to is an estimated life expectancy without the pandemic perhaps obtained using a trend in life expectancy in the pre-pandemic period. In some sense, their analysis shows a lower bound of the impact. Also for 2020, although there is an increase in life expectancy, the actual impact may be negative (if a decrease in LE is smaller than a natural increase in life expectancy). Their main conclusion of a small change in life expectancy may be misleading.
2. I think the decomposition analysis is hard to interpret without a comparator. Would it be possible to include a similar analysis for pre-pandemic period or for year 2020?
3. The finding of no/little correlation between COVID-19 cases and life-expectancy could indicate some other factors may affect the decrease. Please elaborate this in discussion. Related to this point, figure 3 indicate that a slight positive relationship seems driven solely by the two dots with very high changes in LE. It would be nice to discuss what was going on in these prefecture.
4. Having said this, my suggestion is to include a few years of the pre-pandemic period and to assess how a change in LE has been altered before and after the pandemic.

Reviewer 2 ·

Basic reporting

The article was written in professional English which is clear and easy to follow. The tables and figures were informative and self-contained. However, on several occasions, the references were not cited appropriately to support the argument in the main text. Notably, in line 84, while describing the coronavirus situation in Japan, the authors cited a paper focusing exclusively on Italy (Blangiardo et al., 2020). In the paragraph above (lines 75-76), Hanson et al. (2022) and Govarts et al. (2023) were cited to support the indirect effects of Covid-19, but the two papers were not listed in the Reference.

Another confusion occurred in lines 250-251. While discussing the potential effects on neoplasm detection, the authors cited three studies, namely Edamitsu et al. (2022), Santomauro et al. (2021), and Yamasoba et al. (2022), but none of the papers examined utilization patterns during COVID-19 or neoplasm detection in particular. Because increased neoplasm death was indicated as a major factor in the life expectancy change, the lack of sufficient discussion in relation to the literature tends to weaken the validity of the result.

Experimental design

The life expectancy changes were calculated from mortality statistics and population base numbers in 2019-2021. Detailed formulas were presented and sources documenting the methodology were cited. Raw data tables and additional results were provided in the Supplementary Material. Overall, the statistical analysis appears competent and consistent with the standard approach in the literature.

However, because the 2021 data used in the current analysis were preliminary, one way to improve the article is to re-calculate life expectancy changes using updated mortality and exposure-to-risk data for year 2021. This is important because life expectancy could be sensitive to even small changes in the mortality statistics, and the decomposition across causes of death could also differ depending on the final data. These issues should be thoroughly investigated to support the conclusion from the analysis.

Validity of the findings

The strength of the analysis is that all statistical analyses were conducted using open, readily accessible data, making replication viable. However, I would suggest a few additional analyses to further validate the results.

1. calculate life expectancy as well as cause-specific contributions to changes based on updated data

2. in Figure 3, authors plotted life expectancy change against severe cases of COVID-19; what would the relationship look like if life expectancy changes were plotted against COVID-19 mortality per 10,000 population?

3. the authors suggested that delayed detection of neoplasm tumor could be driving the life expectancy change in 2021. One way to assess this hypothesis is to check if life expectancy changes were linked to cancer mortality rates -- for instance, what would the relationship look like if cancer mortality rates were plotted on the x-axis in Figure 3? The mortality rates are published annually in the Cancer Statistics Yearbook, available at https://ganjoho.jp/public/qa_links/report/statistics/2023_en.html

4. more broadly, a large literature has shown reductions in hospital visits and routine healthcare during COVID-19 in 2020-2021. If this is related to delayed detection and treatment of neoplasm tumor, is there a significant relationship if life expectancy changes were plotted against inpatient/outpatient visits in Figure 3? For instance, data on hospital visits by type of beds (available at https://www.e-stat.go.jp/stat-search/files?page=1&layout=datalist&toukei=00450023&tstat=000001030749&cycle=7&tclass1=000001169606&tclass2=000001169614&tclass3=000001169616&tclass4val=0) might be worth exploring, among other statistical sources.

---

## Round 0.2 · accepted · Accept

The authors have satisfactorily addressed all reviewers' comments, and I consider the new comments optional and of no impact on the quality of the manuscript.

Reviewer 1 ·

Basic reporting

I will skip this as I described this before.

Experimental design

This is not a study based on an experiment.

Validity of the findings

No comments.

Additional comments

I am satisfied with the response by authors to my previous comments. Two additional comments.
1. L 235 Supplemental Figure 1 shows 2011 is another deviation from the trend, most likely due to the earthquake occurred in the year. Indeed, the deviation in 2011 looks slightly larger. Please discuss this briefly.
2. L237. A comparison of the magnitude of mortality reductions under the pandemic could provide a useful insight to interpret the impact across countries. Please refer to the size in other countries and compare them with that from the current study.

Reviewer 2 ·

Basic reporting

The reporting style is scientific and professional.

Experimental design

The research question is meaningful. The research methods are well documented and consistent with standard practices in the literature.

Validity of the findings

Multiple datasets are used to confirm and interpret the findings as well as potential mechanisms. Limitations and directions for future work are discussed.

Additional comments

The authors have provided detailed replies to the review. The robustness checks and additional analyses in general support the original findings of the paper. I have no further comments or suggestions.